Retrospective study of BRAFV600E mutation and CT features of papillary thyroid carcinoma

Hong Xiaoquan 1
Li Juxiang 2
Duan Shaoyin 2
You Youkuang 3 yyk577@163.com
1 Department of General Surgery, Zhongshan Hospital of Xiamen University , Xiamen, Fujian , China
2 Department of Medical Imaging, Zhongshan Hospital of Xiamen University , Xiamen, Fujian , China
3 Department of Medical Imaging, Xiamen Xianyue Hospital, Xianyue Hospital Affiliated with Xiamen Medical College, Fujian Psychiatric Center, Fujian Clinical Research Center for Mental Disorders , Xiamen, Fujian , China
Liu Feng
Electronic publication date: 2024 Jan 24
Publication date: 2024
Volume: 12
Electronic Location ID: e16810
Received 2023 Sep 18; Accepted 2023 Dec 29
Copyright: © 2024 Hong et al.
Copyright year: 2024
Copyright holder: Hong et al.
License: This is an open access article distributed under the terms of the Creative Commons Attribution License, which permits unrestricted use, distribution, reproduction and adaptation in any medium and for any purpose provided that it is properly attributed. For attribution, the original author(s), title, publication source (PeerJ) and either DOI or URL of the article must be cited.
License URL: https://creativecommons.org/licenses/by/4.0/

Keywords: Papillary thyroid cancer, BRAFV600E mutation, CT imaging, Clinicopathology, Retrospective study

Funding: The authors received no funding for this work.

==============================
Objective

This study aimed to examine the correlation between BRAFV600E status and computed tomography (CT) imaging characteristics in papillary thyroid carcinoma (PTC) and determine if suspicious CT imaging features could predict BRAFV600E status.

Methods

This retrospective study included patients with pathologically confirmed PTC at the Department of Thyroid Surgery of Zhongshan Hospital, Xiamen University, between July 2020 and June 2022. We compared the clinicopathologic factors and CT findings of nodules with and without the mutation, and the multiple logistical regression test was used to determine independent parameters of the BRAFV600E mutation.

Results

This study included 381 patients with PTC, among them, BRAFV600E mutation was detected in 314 patients (82.4%). Multivariate logistic regression analysis showed that gender (OR = 0.542, 95% CI [0.296–0.993], P = 0.047) and shape (OR = 0.510, 95% CI [0.275–0.944], P = 0.032) were associated with BRAFV600E mutation.

Conclusions

Compared to BRAFV600E mutation-negative, BRAFV600E-positive PTC lesions were more likely to be found in female patients and were characterized by irregular shape. However, the CT imaging finding is not enough to predict BRAFV600E status, but an indication.

Introduction

Papillary thyroid carcinoma (PTC), the most common type among thyroid neoplasms, accounts for about 80% to 95% of all pathological thyroid lesions and is characterized by early lymph node metastasis (Coca-Pelaz et al., 2020; Bai, Kakudo & Jung, 2020). Although PTC prognosis is generally good, with a 95–97% overall 10-year survival rate, approximately 5–10% of patients develop local recurrence and 10–15% of patients have distant metastases, that reduce the 10-year survival rate to 40% (Krajewska et al., 2020; Lowenstein et al., 2019). Due to the fact that traditional prognostic staging systems based on histopathological parameters cannot be used for preoperative assessment, new PTC risk stratification strategies are currently discussed (Grani et al., 2019).

One of the genetic alterations that play a key role in the development of thyroid cancer is the BARFV600E mutation, which is prevalent in PTC (Trovisco, Soares & Sobrinho-Simoes, 2006). This mutation has been previously established as be related to many clinical aspects of PTC. In particular, BRAFV600E mutated PTC exhibits more aggressiveness, tends to infiltrate the peri-thyroid tissue, higher rates of lymph node metastasis, later clinical staging and worse prognosis (Park et al., 2016; Xing, 2007; Elisei et al., 2008). Prior studies have demonstrated that BARFV600E gene mutation is closely related to the clinicopathological features of PTC, tumor recurrence and reduced sensitivity of radioiodine therapy (Xing, 2007). Thus, BRAFV600E genetic testing via ultrasound-guided fine-needle aspiration biopsy (FNAB) has been considered a valid prognostic approach in PTC management, including diagnosis, surgical strategies, selection of radioiodine therapy, and postoperative follow-up (Xing, 2007). Despite the above advantages, fine needle aspiration biopsy (FNAB) remains an invasive test usually scheduled preoperatively to facilitate clinical diagnosis and treatment; performing this test without adequate screening could add to the economic burden of the disease and potentially increase complications, such as repeated invasive FNA examinations and bleeding.

Preoperative utilization of non-invasive imaging techniques to predict BRAFV600E mutation may influence surgical decision-making, particularly for primary medical institutions without technical conditions for FNAB or in cases where patients are not contraindicated for FNAB operation, such as those with poor coagulation and cardiopulmonary function (Grani et al., 2019). Correlation between clinical and ultrasound features and BRAFV600E mutation status in PTC has been previously investigated, but the published results are inconsistent. Some scholars believe that there is no association (Liu et al., 2005; Park et al., 2014; Al-Masri et al., 2021; Hwang et al., 2010), while others suggest that BRAFV600E positivity was associated with most suspicious ultrasound characteristics, including taller-than-wide shape, ill-defined margins, hypoechogenicity, calcifications, and absent halo (Kabaker et al., 2012; Zhang et al., 2017; Khadra et al., 2018). However, the major limitation of ultrasound is its high dependency on the operator’s experience. Computed tomography (CT) three-dimensional reconstruction offers a detailed visualization of thyroid lesions and their adjacent relationships, providing a certain basis for clinicians to assess the disease. To our knowledge, only a few reports have examined the value of CT imaging features in prediction of BRAFV600E mutation status. This study fills a significant gap in the current literature and offers a novel approach to assess the BRAFV600E mutation status.

The objective of this study was to examine the correlation between BRAFV600E mutation and clinicopathological features and CT imaging characteristics in PTC and thereby determine whether suspicious CT imaging characteristics could predict BRAFV600E status.

Materials and Methods

All procedures performed in studies involving human participants were in accordance with the ethical standards of the institutional. Informed consent was waived by Institutional Review Board due to retrospective study characteristics.

Study design and population

This retrospective study was approved by the institutional review board of Zhongshan Hospital affiliated to Xiamen University, and informed consent was waived. The Ethical Approval number was xmzsyyky-ethical-review-NO 2022-185. This retrospective controlled study included patients with surgically and pathologically confirmed PTC, who were treated at the Department of Thyroid Surgery of Zhongshan Hospital, Xiamen University, between July 2020 and June 2022. Inclusion criteria were as follows: (1) PTC confirmed by pathology examination; (2) IQon spectral CT plain and enhanced examinations within 1 month before surgery; (3) maximum diameter of the primary lesion ≥0.5 cm. Exclusion criteria were (1) multiple thyroid carcinomas; (2) significant clavicular artifacts; (3) obvious Hashimoto’s thyroiditis affecting nodal observation; (4) previous thyroid operation; (5) absence of BRAFV600E gene test.

Data collection

Demographic data such as age, gender and pathological findings including extrathyroidal manifestations, lymph node metastasis of the patients were collected.

All patients underwent an ultrasound-guided fine-needle aspiration. Cytology slides from FNA were retrieved for BRAFV600E analysis. Genomic DNA was extracted from aspirated thyroid cells using the ADx DNA mini kit (Amoy) according to the manufacturer’s instructions. Allele-specific amplification blocks polymerase chain reaction (ARMS-PCR) was used to detect the BRAFV600E mutation. A case was considered BRAFV600E positive if the method identified the presence of the mutation, otherwise it was analyzed in the BRAF-negative group.

Image acquisition and analysis

Images were obtained by using a Philips IQon spectral CT scanner. All patients were scanned by using the following parameters: tube current, 148–202 mA; collimation thickness, 0.625 mm; helical pitch 1.14; reconstruction thickness 2 mm and the use of Philip iDose4 level-3 iterative reconstruction algorithms. For contrast material-enhanced scanning, an iodinated nonionic contrast agent (Uvexan 300) was injected through the right ulnar vein at 3 mL/s by using an automated injector. The patient was scanned from the oropharynx to the level of the superior edge of the aortic arc. ROI was placed in the aortic arch position, and the scan was automatically triggered. The CT value of the aortic arch reached 150 HU to start the arterial phase, and the arterial phase was scanned for 30s before entering the venous phase. Two radiologists with 10 and 8 years of work experience retrospectively analyzed the imaging data independent of each other, including the longest diameter of the nodule (d ≤ 10 mm, >10 mm), shape (regular or irregular), margin (well or ill defined), area of marginal contact (≤1/4, 1/4 to <1/2 and ≥1/2) (Zhan et al., 2012), calcification (positive or negative), difference in density between arterial and plain scan, and difference in density between venous and plain scan. When the diagnosis was inconsistent, disagreement was resolved by consensus.

Statistical analysis

Data was analyzed using statistical software SPSS 26.0 (IBM Corp., Armonk, NY, USA). The continuous data not conforming to the normal distribution were presented as medians (interquartile interval) (M (Q25-Q75)) and analyzed using the Mann-Whitney U test. The categorical data were expressed as n (%) and analyzed using the chi-square test. Multivariate logistic regression analysis was used to evaluate associations between BRAFV600E mutation status and risk factors included in univariate analysis. Odds ratios and their relative 95% confidence intervals were also calculated to determine the relevance of all potential predictors of outcome. Two-tailed P-values < 0.05 were considered statistically significant.

Results

A total of 580 patients were enrolled. Finally, a total of 381 PTC patients were included, with a median age of 41 years (range 15–71 years). There were 314 cases (82.4%) in the BRAFV600E mutation positive group (Fig. 1) and 67 cases (17.6%) in negative group (Fig. 2). Compare to the BRAFV600E mutation negative group, the proportion of females was slightly higher in the BRAFV600E mutation positive group (P = 0.036), otherwise two groups were comparable in age, extrathyroidal manifestations or lymph node metastasis (all P > 0.05) (Table 1).

Figure 1 A 39-year-old woman with BRAFV600E mutation-positive PTC.

(A) Unenhanced CT shows an irregularly shaped, poorly defined margin, and microcalcification-containing lesion with a CT value of 62HU was observed. (B and C) In the contrast-enhanced phase image, the lesion exhibits mild to moderate enhancement, with CT values of 128HU and 106HU, respectively. The difference in density between the arterial and plain scan, as well as the venous and plain scan, is 66HU and 44HU, respectively. (D) The amplification plot of BRAFV600E demonstrates the BRAFV600E mutation type. The middle curve, representative of the BRAFV600E sample, is situated between the upper reference curve, indicating BRAFV600E positivity, and the horizontal lower line, suggesting BRAFV600E negativity.

Figure 2 A 22-year-old man with BRAFV600E mutation-negative PTC.

(A) An unenhanced CT scan reveals a nodule with a regular shape, well-defined margins, and microcalcifications. The area of marginal contact ranges from 1/4 to less than 1/2, with a CT value of 35HU. (B and C) In the contrast-enhanced phase image, the lesion exhibits significant enhancement, with CT values of 116HU and 148HU, respectively. The differences in density between the arterial and plain scans, as well as the venous and plain scans, are 81HU and 113HU, respectively. (D) The BRAFV600E amplification plot demonstrates a wild-type BRAFV600E. The middle curve, representing the BRAFV600E sample, overlaps with the horizontal lower reference line, indicating a BRAFV600E-negative result.

Table 1 Papillary thyroid carcinoma patients with and without BRAFV600E mutation.

Clinicopathological features	BRAFV600E mutation	χ2	P	
Positive (n = 314)	Negative (n = 67)	
Gender			4.379	0.036	
Female	232 (73.89)	41 (61.19)			
Male	82 (26.11)	26 (38.81)	
Age (years)			1.543	0.214	
≦45	195 (62.10)	47 (70.15)			
>45	119 (37.90)	20 (29.85)	
Extrathyroidal manifestations	169 (53.82)	29 (43.28)	2.457	0.117	
Lymph node metastasis	192 (61.15)	42 (62.69)	0.055	0.814	

There were significant differences in CT features depending on the presence or absence of BRAFV600E mutation. In particular, irregular shape of the lesion was more common in the BRAFV600E mutation positive group (50.32%, compared to 29.85% in the negative group; P = 0.002). Well-defined margin was observed less often in the BRAFV600E mutation positive group (46.50%, compared to 62.69% in the negative group; P = 0.016). The differences in density between arterial and plain scan, venous and plain scan were statistically significant between the BRAFV600E mutation positive and negative group (52.40 (32.13–73.43) vs. 65.60 (40.30–83.70), P = 0.008 and 47.90 (33.10–62.03) vs. 53.10 (43.30–67.70), P = 0.016). None of the other CT features such as the longest diameter of the nodule, area of marginal contact, calcification were significantly associated with BRAFV600E mutation in PTC patients (P > 0.05) (Table 2).

Table 2 CT imaging features of papillary thyroid carcinoma patients with BRAFV600E mutation.

CT image features	BRAFV600E mutation	χ 2 /Z	P	
Positive (n = 314)	Negative (n = 67)	
Longest diameter (mm)			2.029	0.154	
≤10	166 (52.87)	29 (43.28)			
>10	148 (47.13)	38 (56.72)			
Shape			9.293	0.002	
Regular	156 (49.68)	47 (70.15)			
Irregular	158 (50.32)	20 (29.85)			
Calcification	128 (40.76)	35 (52.23)	2.970	0.085	
Aarea of marginal contact			5.194	0.256	
≤1/4	93 (29.62)	13 (19.40)			
1/4~1/2	93 (29.62)	17 (25.37)			
≥1/2	69 (21.97)	21 (31.34)			
Margin			5.790	0.016	
Well-defined	146 (46.50)	42 (62.69)			
Ill-defined	168 (53.50)	25 (37.31)			
Difference in density between arterial and plain scan (HU)	52.40 (32.13–73.43)	65.60 (40.30–83.70)	−2.761	0.008	
Difference in density between venous and plain scan (HU)	47.90 (33.10–62.03)	53.10 (43.30–67.70)	−2.410	0.016	

The above statistically significant factors on univariate analysis were incorporated into multivariate logistic regression analysis, which showed that BRAFV600E mutation was associated with gender (OR = 0.542, 95% CI [0.296–0.993], P = 0.047) and shape (OR = 0.510, 95% CI [0.275–0.944], P = 0.032) (Table 3).

Table 3 Multivariate analysis of BRAFV600E mutations in PTC and clinical imaging features.

	OR	95% CI	P	
Gender	0.542	[0.296–0.993]	0.047	
Age > 45	1.421	[0.781–2.588]	0.250	
Longest diameter	0.702	[0.402–1.225]	0.213	
Shape	0.510	[0.275–0.944]	0.032	
Margin	0.637	[0.352–1.154]	0.137	
Difference in density between arterial and plain scan	1.009	[0.996–1.022]	0.174	
Difference in density between venous and plain scan	1.010	[0.992–1.027]	0.274	

Discussion

In this study, we have demonstrated that there were some correlations between BRAFV600E status in patients with PTC and CT features including irregular shape and the difference in density between arterial/venous and plain scans. In addition, the results also showed a higher proportion of females in the BRAFV600E mutation positive group, but comparable results regarding age or tumor size.

Irregular shape and ill-defined margin are common malignant signs of tumors, due to the absence of capsule and the uncontrolled and irregular proliferation of tumor tissue. Our results indicate that the BRAFV600E mutation-positive group more often had irregular shape in CT images (OR = 0.510, 95% CI [0.275–0.944], P = 0.032) and a more significant correlation between the BRAFV600E mutation and irregular shape could be found on univariate analysis (P = 0.002). Although the statistical difference was not significant (P = 0. 032), we suggest that an irregular shape of PTC can be used as an indication of an aggressive tendency in tumor growth, which is associated with BRAFV600E mutation. This finding may contribute to search of affordable screening methods that ease the burden of FNAB in PTC management, and might be used during the future clinical work and research.

We also observed a significant correlation between the BRAFV600E mutation and the difference in density between arterial/venous and plain scans, based on univariate analysis. However, this correlation disappeared when we performed a multivariate analysis, adjusting for age, gender, longest diameter, shape, and margin. The difference in density between arterial/venous and plain scans of PTC is partly attributed to the degree of iodine absorption in addition to blood supply. Indeed, previous studies have demonstrated that the BRAFV600E mutation was associated with a decrease in radioiodine uptake and loss of radioiodine avidity (Xing, 2007; Durante et al., 2007). Therefore, further research on a larger scale will be necessary to appreciate the additional value of the difference in density between arterial/venous and plain scans in terms of the BRAFV600E mutation.

Relationships between BRAFV600E mutation and traditionally discussed high-risk factors of thyroid cancer, such as age, gender and tumor size are controversial. Some studies believe that there is no association (Liu et al., 2005; Park et al., 2014; Al-Masri et al., 2021), while others report that the BRAFV600E mutation patients are more often male and have more nodules at older age (Park et al., 2014; Zhang et al., 2017). Numerous investigations have demonstrated a significantly higher incidence of thyroid cancer in women compared to men and the previously reported presence of estrogen receptor in thyroid glands (Miki et al., 1990; Lewy-Trenda, 2002) suggests that estrogen plays an important role in development and progression of thyroid neoplasms due to the effect of sex hormones (Rahbari, Zhang & Kebebew, 2010). Prior studies indicated a significantly associated expression of ER with gender in PTC tissue (Wang et al., 2018; Fan et al., 2015), which could explain the higher frequency of females with BRAFV600E mutation. However, gender peculiarities of the mutation might not be ruled out and should be closer investigated in the future.

Various oncogenic mutations have been identified in thyroid cancer (Nikiforov, 2002; Davies et al., 2002). The BRAFV600E mutation, the most common mutation in patients with PTC, persistently activates the RAS/RAF/MEK/MAPK pathway and promotes thyroid cell proliferation. Previous studies have shown that BRAFV600E mutation significantly increase the risk of extrathyroidal extension, lymph node metastasis and advance TNM stage (Xing, 2007; Elisei et al., 2008; Park et al., 2016; Abdullah et al., 2019). Thus it is seen as an independent predictive factor for poor prognosis in PTC (Kabaker et al., 2012; Khadra et al., 2018; Kim et al., 2012). However, our study found no significant relationship between the BRAFV600E mutation and any clinicopathologic features, including extrathyroidal manifestation and lymph node metastasis, which was consistent with the previously reported findings (Liu et al., 2005; Gouveia et al., 2013; Ito et al., 2009). We consider that this finding might be attributed to the inclusion of 51% microcarcinoma and exclusion of patients with multiple thyroid carcinomas.

This study has several limitations. Firstly, retrospective single-center study design might have led to some selection bias. Secondly, pathological types of PTC, such as classic, high columnar cell subtype and follicular papillary thyroid carcinoma were not separately analyzed, and the differences between subtypes might have affected the specificity of CT scan results. Finally, the postoperative follow-up data was not analyzed in the scope of this study. Further prospective multicenter studies are needed to analyze the relationship between CT features, BRAFV600E and the prognosis of PTC.

Conclusions

In conclusion, BRAFV600E mutation-positive PTC lesions were characterized by irregular shape and were more likely to be found in female patients compared to BRAFV600E mutation-negative PTC. However, the CT imaging finding is not enough to predict BRAFV600E status, but an indication.

Supplemental Information

Supplemental Information 1 Raw data.

Click here for additional data file.

Additional Information and Declarations

Competing Interests

Author Contributions

Human Ethics

Data Availability

The authors declare that they have no competing interests.

Xiaoquan Hong conceived and designed the experiments, performed the experiments, analyzed the data, prepared figures and/or tables, authored or reviewed drafts of the article, and approved the final draft.

Juxiang Li conceived and designed the experiments, performed the experiments, analyzed the data, prepared figures and/or tables, authored or reviewed drafts of the article, and approved the final draft.

Shaoyin Duan performed the experiments, analyzed the data, authored or reviewed drafts of the article, and approved the final draft.

Youkuang You analyzed the data, authored or reviewed drafts of the article, and approved the final draft.

The following information was supplied relating to ethical approvals (i.e., approving body and any reference numbers):

The institutional Review Board of Zhongshan affiliated to Xiamen university granted Ethical approval to carry out the study within its facilities (Ethical Application Ref: xmzsyyky-ethical-review-NO 2022-185).

The following information was supplied regarding data availability:

The raw measurements are available in the Supplemental Files 1. The raw data shows all BRAFV600E mutation status, CT imaging and clinicopathology of all 381 PTC patients.

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
