# Peer review of "Retrospective study of BRAFV600E mutation and CT features of papillary thyroid carcinoma"

_PeerJ, doi:10.7717/peerj.16810_

## Round 0.1 · original submission · Major Revisions

The authors should carefully address the concerns.

**Language Note:** The review process has identified that the English language must be improved. PeerJ can provide language editing services - please contact us at copyediting@peerj.com for pricing (be sure to provide your manuscript number and title). Alternatively, you should make your own arrangements to improve the language quality and provide details in your response letter. – PeerJ Staff

·

Basic reporting

no comment

Experimental design

no comment

Validity of the findings

no comment

Additional comments

It is crucial and meaningful to determine the genotype of papillary thyroid carcinoma. In this study, CT image features were used to characterize the genotype of PTC, which provided a new perspective for future research. However, several points need to be addressed in a revision of the manuscript.

Introduction:

BARFV600E mutation or not, which plays a key role in the development of thyroid cancer. However, it might be beneficial to outline more clearly why the CT image features were selected. After all, CT is a radiation examination, and the thyroid is relatively sensitive to radiation. As a minor suggestion, you might consider highlighting how your study fills a gap in the current literature or offers a novel method, just because less researches are not enough.

Materials & Methods:
Image acquisition and analysis section, the image reconstruction algorithm and iterative algorithm should be specified, because it has a great impact on the image quality.

For the image features analysis, diagnostic consistency should be checked. Or, what should be done when the diagnosis is inconsistent?
Results:
Lacking of explanation on the process of multivariate logistic regression.

Lacking of the statistical value corresponding to the p value.

“extrathyroid manifestations or lymph node metastasis” were sudden mentioned in the results. What does extrathyroid manifestations include? Which lymph nodes does lymph node metastasis refer to?

Discussion:
The first paragraph of discussion is not conclusive and straight forward. An introductory paragraph that briefly summarizes the results would be helpful at first.

In the discussion, the key points should be focused on the research results. The association of irregular shape and female and BARFV600E mutation should be carefully discussed.

The related sonography studies are not relevant to this study. There are too many content related to sonography studies, it might be briefly summarized

Tables and figures:
As a minor suggestion, a representative case should be displayed. It might be helpful to understand the relationship between CT image features and BARFV600E mutation.

Reviewer 2 ·

Basic reporting

"This paper discusses the relationship between the BRAFV600E mutation and CT features of papillary thyroid carcinoma to predict the presence of the BRAFV600E mutation based on the characteristic areas of CT images. While CT images alone may not be sufficient to make definitive predictions, they can still exhibit a certain level of correlation. Generally speaking, this manuscript is well-written, clear, and in proper language usage. The references used generally meet the requirements. The chart is displayed clearly, and most of the data is accurate.

Experimental design

While research has reported that the BRAFV600E mutation cannot be used alone as an independent predictive factor in PTC patients, it is prognostically valuable when integrated within the context of other clinical and pathological risk factors. Based on this hypothesis, this paper discusses the interaction between physical marketing and these factors. However, it's worth noting that the sample size of the negative group in this research is somewhat small. There is substantial ethical support for this study, and the statistical methods employed are correct.

Validity of the findings

There are still some areas in this article that require revision and need to be addressed.

1 Research reports have shown that papillary thyroid carcinoma is related to several genes, including BRAFV600E, BRAFK601E, NRAS, HRAS, KRAS, TERT, and HLA-G. In this paper, BRAF status is chosen as the main point of discussion. However, in Part 1 of the discussion, the description is insufficient. It is recommended to include additional discussion points.
2 The article contains multiple formatting errors, and it is advisable to correct them.

---

## Round 0.2 · Minor Revisions

Some minor comments must be addressed.

·

Basic reporting

no comment

Experimental design

no comment

Validity of the findings

no comment

Additional comments

The authors have addressed most of the relevant questions in the revision. However, some minor issues should be focused

The first paragraph of discussion is still not conclusive and straight forward. Please considering deleting line 142 to 148. The statement “This findings may contribute to the ongoing search of affordable screening methods that ease the burden of FNAB in PTC management, and might be used during the planning and implementation of the future clinical studies as well as in the everyday clinical practice.” should be more briefly.

Reviewer 2 ·

Basic reporting

no comment

Experimental design

no comment

Validity of the findings

no comment

Additional comments

no comment

---

## Round 0.3 · accepted · Accept

The manuscript can be accepted now!

·

Basic reporting

no comment

Experimental design

no comment

Validity of the findings

no comment

Additional comments

no comment